# Invariance & Causal Representation Learning: Prospects and Limitations

**Simon Bing**                      *bing@campus.tu-berlin.de*
*Technische Universität Berlin*

**Tom Hochsprung**
*German Aerospace Center (DLR), Institute of Data Science*
*Technische Universität Berlin*

**Jonas Wahl**
*Technische Universität Berlin*
*German Aerospace Center (DLR), Institute of Data Science*

**Urmi Ninad**
*Technische Universität Berlin*
*German Aerospace Center (DLR), Institute of Data Science*

**Jakob Runge**
*ScaDS.AI Dresden/Leipzig, TU Dresden*
*German Aerospace Center (DLR), Institute of Data Science*
*Technische Universität Berlin*

**Reviewed on OpenReview:** *https://openreview.net/forum?id=lpOC6s4BcM*

## Abstract

Learning causal representations without assumptions is known to be fundamentally impossible, thus establishing the need for suitable inductive biases. At the same time, the invariance of causal mechanisms has emerged as a promising principle to address the challenge of out-of-distribution prediction which machine learning models face. In this work, we explore this invariance principle as a candidate assumption to achieve identifiability of causal representations. While invariance has been utilized for inference in settings where the causal variables are observed, theoretical insights of this principle in the context of causal representation learning are largely missing. We assay the connection between invariance and causal representation learning by establishing impossibility results which show that invariance alone is insufficient to identify latent causal variables. Together with practical considerations, we use our results to reflect generally on the commonly used notion of identifiability in causal representation learning and potential adaptations of this goal moving forward.

## 1 Introduction

Inferring high-level causal variables from low-level measurements is a problem garnering increased attention in fields interested in understanding epiphenomena that cannot be directly measured and where controlled experiments are not possible due to practical, economical or ethical considerations, for instance in healthcare (Johansson et al., 2022), biology (Lopez et al., 2023) or climate science (Tibau et al., 2022). This problem of causal representation learning (Schölkopf et al., 2021) has been shown to be fundamentally underconstrained (Locatello et al., 2019), leading to various approaches exploring which assumptions lead to algorithms that identify the latent causal variables.

Recent works either restrict the underlying causal model (Buchholz et al., 2024; Lachapelle et al., 2024), the transformation causal variables undergo (Ahuja et al., 2023; Lachapelle et al., 2023), or both (Squires et al., 2023). They include interventional or counterfactual data (Ahuja et al., 2023; Zhang et al., 2023; Squires et al., 2023; Buchholz et al., 2024; Bing et al., 2024; Brehmer et al., 2022), use supervisory signals such as time structure (Hyvärinen & Morioka, 2017; Hälvä & Hyvärinen, 2020; Yao et al., 2021) or knowledge of intervention targets (Lippe et al., 2022b;a).

We explore another type of inductive bias for achieving identifiability of causal representations, namely the invariance of causal mechanisms (Peters et al., 2017). First shown by Haavelmo (1944), causal variables lead to predictive models that are invariant under interventions, and since causal representation learning is tasked with recovering precisely these variables, we investigate if and to which degree the principle of invariance can be used as a signal to recover latent causal variables from observations.

While invariance has been used for causal inference (Peters et al., 2016; Bühlmann, 2018; Meinshausen, 2018), none of these works considers the setting where we only have access to observations that are related to the underlying causal variables by some unknown transformation. To the best of our knowledge, we present the first theoretical results pertaining to the identifiability of causal representations using the invariance principle.

Our contributions are summarized as follows:

- Drawing on the link between distributional robustness and causality, we formalize the setting in which we study the connection between invariance and causal representation learning.

- We establish *impossibility* results proving the necessity of additional assumptions to achieve identifiability.

- Based on these impossibility results and practical considerations, we contemplate further constraints and map out future work towards identifiable algorithms based on invariance.

## 2 Problem setting

Intuitively, our problem setting can be motivated by considering the prediction problem with a target $Y$ and observations $\mathbf{X}$ in multiple environments. We assume that there is an underlying causal representation of the observations—denoted by $\mathbf{Z}$—whose constituents are causes of the prediction target $Y$ and interact with each other through a structural causal model (SCM). This representation $\mathbf{Z}$ is what we are interested in finding.

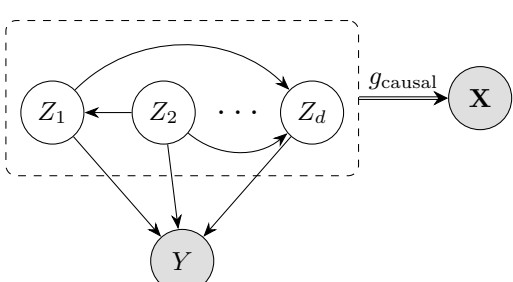

Figure 1: We consider an SCM with variables $(Z_1, \ldots, Z_d)$ and $Y$. Observed variables are represented by shaded nodes, indicating that we do not observe $(Z_1, \ldots, Z_d)$, but only their transformation $\mathbf{X} = g_{\text{causal}}(\mathbf{Z})$.

**Notation.** We denote scalar variables in normal face $(x)$ and use bold face for vector-valued variables $(\mathbf{x})$. We capitalize random variables $(Y)$, and write the values they take in lower case $(y)$. Matrices are denoted capitalized and bold $(\mathbf{M})$ and are explicitly introduced to avoid confusion with vector-valued random variables. The sequence of integers from 1 to $n$ is denoted with $[n]$.

**Data generating process.** Let $(Z_1, \ldots, Z_{d+1})$ denote a set of random variables. W.l.o.g. we call $Z_{d+1}$ the *target* variable and rename it $Y$, denoting the remaining $d$ variables with $\mathbf{Z} = (Z_1, \ldots, Z_d)$. Assume an SCM defined over the random vector $(\mathbf{Z}, Y)$ inducing the joint distribution $P(\mathbf{Z}, Y)$ over $(\mathbf{Z}, Y)$. The graph $\mathcal{G}$ induced by this SCM is assumed to be acyclic. Assume $Y$ is not a parent of any $\mathbf{Z}$. Since $P(\mathbf{Z}, Y)$ is induced by an SCM, it factorizes as

$$P(\mathbf{Z}, Y) = \prod_{i=1}^{d+1} P(Z_i \mid Z_{\text{Pa}_i}), \tag{1}$$

where $\text{Pa}_i \subset [d+1] \setminus i$ denotes the parents of variable $Z_i$. We refer to (Pearl, 2009) for an in-depth definition of SCMs.

Beyond the additive noise assumption we do not place any parametric constraints on the causal mechanism of each variable, i.e. each structural equation can be written as $Z_i := f_i(Z_{\text{Pa}_i}) + \varepsilon_i$, where $\varepsilon_i$ with $i \in [d+1]$ denote the exogenous, independent noise terms, which are assumed to have zero mean. Since the causal mechanism of the target $Y$ is of particular interest in a predictive setting, we denote it with $f_{\text{causal}} := f_{d+1}$.

We do not directly measure $\mathbf{Z}$ and only assume to observe $\mathbf{X} \in \mathbb{R}^p$, where $\mathbf{X} = g_{\text{causal}}(\mathbf{Z})$ is a transformation of the causal variables $\mathbf{Z}$ by the injective, deterministic (and potentially nonlinear) function $g_{\text{causal}}$. We assume $p \geq d$. Notice that $Y$ is *not* transformed by $g_{\text{causal}}$; we assume that the target variable is directly observed. The set of observed variables is therefore denoted by $(\mathbf{X}, Y)$. Fig. 1 depicts a graphical representation of this data generating process.

We assume to observe $(\mathbf{X}, Y)$ across multiple environments, where subsets $S \subseteq [d]$ of the underlying latent variables $\mathbf{Z}$ have undergone an intervention in each environment. We model interventions with do-interventions (Pearl, 2009) (also called hard interventions), which set the structural equations of the variables that are targeted by an intervention to constant values, allowing us to write

$$Z_j := a_j \quad \text{for } j \in S, \tag{2}$$

where $\mathbf{a} \in \mathbb{R}^{|S|}$. Each intervention on the subset of variables $S$ with value $\mathbf{a}$ induces a new distribution over $(\mathbf{Z}, Y)$ which, following the notation introduced by Meinshausen (2018), is denoted by $P_{\mathbf{a},S}^{(\text{do})}$. We assume $Y$ is never among the set of intervention targets and the mixing function $g_{\text{causal}}$ does not change between environments.

**Objective.** Although we explicitly define a target variable, we stress that our main goal is not to learn a good predictive model of $Y$, but rather to recover the latent variables $\mathbf{Z}$, i.e. we focus on the representation learning aspect of the setting described above. Our aim is to probe how the auxiliary task of predicting $Y$ from transformations of latent variables in multiple environments can help in recovering the unobserved causal variables. Our formal objective is to invert the mixing function $g_{\text{causal}}$ in order to recover the latent variables $\mathbf{Z}$ from observations $\mathbf{X}$. Since latent variables that are equal to the ground truth up to permutation and element-wise rescaling can give rise to the same observations $\mathbf{X}$ (Zhang et al., 2023), we define an equivalence relation over this class of latents. Consequently, we define recovering the latent causal variables up to this equivalence as our notion of identifiability. Equivalent representations are referred to as *causally disentangled*.

**Definition 1** (Causally Disentangled Representations, Khemakhem et al. (2020); Lachapelle et al. (2022))**.** *A learned representation $\hat{\mathbf{Z}}$ is causally disentangled w.r.t. the ground truth representation $\mathbf{Z}$ if there exists an invertible diagonal matrix $\mathbf{D}$ and a permutation matrix $\mathbf{P}$ s.t. $\hat{\mathbf{Z}} = \mathbf{DPZ}$ almost surely.*

## 3 Invariance for causal structure learning

The connection between (predictive) invariance and causality is long established: Haavelmo (1944) was the first to formalize that a model which predicts a target from its direct causes is invariant under interventions on any other covariates of the system. In the language of SCMs this means that the conditional distribution $P(Y|\mathbf{Z}_{\text{Pa}_Y})$ remains invariant under any interventions on $\mathbf{Z}$. This principle is also referred to as autonomy (Aldrich, 1989), modularity (Pearl, 2009) and independence of cause and mechanism (Peters et al., 2017).

More recently, the opposite direction has been explored, namely how invariance can be leveraged as a signal to infer causal structures and mechanisms. Peters et al. (2016) pioneered this approach by exploiting the principle of invariance of causal mechanisms to infer the direct causes of a target $Y$, assuming direct observations of the causal variables $\mathbf{Z}$. A particularly interesting line of works draws a connection between distributional robustness and causality (Meinshausen, 2018; Bühlmann, 2018; Rojas-Carulla et al., 2018) by considering the problem

$$\min_f \sup_{Q \in \mathcal{Q}} \mathbb{E}_{(\mathbf{Z},Y) \sim Q} \left[ (Y - f(\mathbf{Z}))^2 \right], \tag{3}$$

where $\mathcal{Q}$ denotes some set of interventional distributions. Christiansen et al. (2022) investigate under which choices of $\mathcal{Q}$ the function $f_{\text{causal}}$ remains a solution and Meinshausen (2018) (see also Rojas-Carulla et al. (2018)) states that $f_{\text{causal}}$ is the *unique* solution to this problem when $f_{\text{causal}}$ is linear and $\mathcal{Q}$ is the set of interventions on *all* variables except $Y$, with arbitrary strength. None of the mentioned works consider settings where $\mathbf{Z}$ is latent.

As a first step towards our main theoretical findings, we extend the results of Meinshausen (2018) by showing that the causal mechanism of the target $f_{\text{causal}}$ is the unique solution to Eq. (3) for general nonlinear $f_{\text{causal}}$ (with additive noise) when the set of interventions $\mathcal{Q}$ contains interventions on all covariates $\mathbf{Z}$, or on all subsets of covariates $\mathbf{Z}$. Assuming interventions on all variables or all subsets of variables in $\mathbf{Z}$ can be understood as a diversity condition on the observed environments.

**Lemma 1.** *Assume the general SCM presented in Section 2 as the data generating process and consider the robust optimization problem described by Eq. (3). Let $\mathcal{Q}^{(\text{do})} := \left\{ P_{\mathbf{a},[d]}^{(\text{do})}; \; \mathbf{a} \in \mathbb{R}^d \right\}$, i.e. the set of distributions resulting from do-interventions on* all *variables, except $Y$, with arbitrary strength. Then, the causal mechanism of the target $Y$, $f_{\text{causal}}$, is the unique optimizer of Eq. (3), i.e.*

$$f_{\text{causal}} = \arg\min_f \sup_{Q \in \mathcal{Q}^{(\text{do})}} \mathbb{E}_{(\mathbf{Z},Y) \sim Q} \left[ (Y - f(\mathbf{Z}))^2 \right].$$

*The same results holds true if one replaces $\mathcal{Q}^{(\text{do})}$ with the set of all do-interventions $\mathcal{Q}'^{(\text{do})}$ on all possible subsets of variables, except $Y$, with arbitrary strength.*

*Proof.* Suppose that $f \neq f_{\text{causal}}$, which implies that there exists at least one $\mathbf{z} = (z_1, \ldots, z_d)$ such that $f(\mathbf{z}) \neq f_{\text{causal}}(\mathbf{z})$. Now, consider the decomposition of the objective

$$
\begin{aligned}
\mathbb{E}_Q \left[ (Y - f(\mathbf{Z}))^2 \right] &= \mathbb{E}_Q \left[ (f_{\text{causal}}(\mathbf{Z}) - f(\mathbf{Z}) + \varepsilon_Y)^2 \right] \\
&= \mathbb{E}_Q \left[ (f_{\text{causal}}(\mathbf{Z}) - f(\mathbf{Z}))^2 \right] + \mathbb{E}_Q \left[ \varepsilon_Y^2 \right] + 2\mathbb{E}_Q \left[ (f_{\text{causal}}(\mathbf{Z}) - f(\mathbf{Z}))\varepsilon_Y \right].
\end{aligned}
\tag{4}
$$

For any interventional distribution $Q \in \mathcal{Q}^{(\text{do})}$ we notice

$$2\mathbb{E}_Q \left[ (f_{\text{causal}}(\mathbf{Z}) - f(\mathbf{Z}))\varepsilon_Y \right] = 2(f_{\text{causal}}(\mathbf{a}) - f(\mathbf{a}))\mathbb{E}_Q \left[ \varepsilon_Y \right] = 0,$$

i.e. the last term in the decomposition in Eq. (4) vanishes. Here we used that $\varepsilon_Y$ has expectation zero. Next, we focus on the first term in the decomposition. We want to find an interventional distribution $Q$ s.t.

$$\mathbb{E}_Q \left[ (f_{\text{causal}}(\mathbf{Z}) - f(\mathbf{Z}))^2 \right] > 0.$$

To do so, we simply choose $\mathbf{a}$ such that $f_{\text{causal}}(\mathbf{a}) \neq f(\mathbf{a})$. We know such a choice of $\mathbf{a}$ exists since $f_{\text{causal}}$ and $f$ are not equal by assumption. For this particular intervention we have

$$\mathbb{E}_Q \left[ (f_{\text{causal}}(\mathbf{Z}) - f(\mathbf{Z}))^2 \right] = \mathbb{E}_Q \left[ (f_{\text{causal}}(\mathbf{a}) - f(\mathbf{a}))^2 \right] = (f_{\text{causal}}(\mathbf{a}) - f(\mathbf{a}))^2 > 0.$$

Thus,

$$\mathbb{E}_Q[(Y - f(\mathbf{Z}))^2] > \mathbb{E}_Q[\varepsilon_Y^2] = Var(\varepsilon_Y),$$

for any $Q \in \mathcal{Q}^{(\text{do})}$. Therefore, the supremum over $\mathcal{Q}^{(\text{do})}$ of the l.h.s. of Eq. (4), and since $\mathcal{Q}^{(\text{do})} \subseteq \mathcal{Q}'^{(\text{do})}$, the supremum over $\mathcal{Q}'^{(\text{do})}$ is also strictly larger than $Var(\varepsilon_Y)$. In contrast, for $f_{\text{causal}} = f$ and any $Q \in \mathcal{Q}'^{(\text{do})}$,

$$\mathbb{E}_Q[(Y - f(\mathbf{Z}))^2] = \mathbb{E}_Q[\varepsilon_Y^2] = Var(\varepsilon_Y),$$

which thus also holds for the supremum over $\mathcal{Q}^{(\text{do})}$ and the supremum over $\mathcal{Q}'^{(\text{do})}$, and we conclude that $f_{\text{causal}}$ is the unique optimizer of Eq. (3). $\qquad\square$

As we are interested in establishing identifiability results, this uniqueness result provides a potentially fruitful starting point: if the unique solution to this optimization problem is related to the underlying causal model, perhaps we can introduce an adaptation of it that yields the causal representation as the unique solution. Notice that Lemma 1 assumes direct access to the variables $\mathbf{Z}$, while our problem setting of interest is characterized by the central assumption of only observing transformations $\mathbf{X} = g_{\text{causal}}(\mathbf{Z})$ of the underlying causal variables. We investigate the implications of this transformation in the next section.

# 4 Invariance for causal representation learning

To finally arrive at the problem setting we are interested in, we introduce the representation function $g$ to the optimization problem presented in Eq. (3), recalling that $\mathbf{X} = g_{\text{causal}}(\mathbf{Z})$. Now, consider the extended problem

$$\min_{f,g} \sup_{Q \in \mathcal{Q}} \mathbb{E}_{(\mathbf{Z},Y) \sim Q} \left[ \left( Y - f(g^{-1}(\mathbf{X})) \right)^2 \right]. \tag{5}$$

Notice that the expectation is still taken over the joint distribution of $(\mathbf{Z}, Y)$, i.e. the different environments which we consider still arise from interventions on the underlying latent variables $\mathbf{Z}$.

If solving Eq. (3) allows us to uniquely recover $f_{\text{causal}}$, can solving the extended problem in Eq. (5) allow us to draw similar conclusions about $g_{\text{causal}}$?

## 4.1 Causal mechanism and representation are jointly unique

As a first uniqueness result, we show that the joint function composed of the causal mechanism $f_{\text{causal}}$ and the inverse of the ground-truth representation function $g_{\text{causal}}^{-1}$ is the unique solution to Eq. (5).

**Lemma 2.** *Assume the data generating process in Section 2 and consider the optimization problem described in Eq. (5). Let $\mathcal{Q}^{(\text{do})} := \left\{ P_{\mathbf{a},[d]}^{(\text{do})}; \mathbf{a} \in \mathbb{R}^d \right\}$, i.e. the set of do-interventions on* all *underlying variables $\mathbf{Z}$, except $Y$, with arbitrary strength. Define $h := f \circ g^{-1} : \mathbb{R}^p \to \mathbb{R}$ and let $\text{Im}(\cdot)$ denote the image of a function. Then, the composed function $h_{\text{causal}} := (f_{\text{causal}} \circ g_{\text{causal}}^{-1})$ is the unique optimizer of Eq. (5) on $\text{Im}(g_{\text{causal}})$, i.e.*

$$h_{\text{causal}} = (f_{\text{causal}} \circ g_{\text{causal}}^{-1}) \in \arg \min_{h} \sup_{Q \in \mathcal{Q}^{(\text{do})}} \mathbb{E}_{(\mathbf{Z},Y) \sim Q} \left[ (Y - h(\mathbf{X}))^2 \right],$$

*and any other minimizer $h'$ satisfies $h' = h_{\text{causal}}$ on $\text{Im}(g_{\text{causal}})$.*

*The same results holds true if one replaces $\mathcal{Q}^{(\text{do})}$ with the set of all do-interventions $\mathcal{Q}'^{(\text{do})}$ on all possible subsets of variables, except $Y$, with arbitrary strength.*

To prove Lemma 2, we need to find a $\mathbf{b} \in \mathbb{R}^p$ s.t. $h(\mathbf{b}) \neq h_{\text{causal}}(\mathbf{b})$. Notice that $h_{\text{causal}}$ takes as argument $\mathbf{X} = g_{\text{causal}}(\mathbf{Z})$ while we can only intervene directly on $\mathbf{Z}$. Thus, we can only find $\mathbf{b} \in \text{Im}(g_{\text{causal}})$ and consequently the statement $h(\mathbf{b}) \neq h_{\text{causal}}(\mathbf{b})$ only holds on the image of $g_{\text{causal}}$. Taking this into account, the proof (presented in Appendix A) follows the same structure as the proof of Lemma 1.

## 4.2 Overparametrization of the unconstrained setting

While the uniqueness result of Lemma 2 regarding the composed function $(f_{\text{causal}} \circ g_{\text{causal}}^{-1})$ is a first promising direction in relating the solution of a distributional robustness problem to inverting the representation function $g_{\text{causal}}$, we quickly see that this uniqueness result does not yet constrain the individual components $f$ and $g$ of the solution enough.

**Theorem 1.** *Consider the data generation process presented in Section 2. Without additional assumptions, the distributional robustness problem described by Eq. (5) does not suffice to identify the underlying causal variables up to the equivalence class detailed in Definition 1.*

*Proof.* While $(f_{\text{causal}} \circ g_{\text{causal}}^{-1})$ has been shown to be the unique solution to the optimization problem described by Eq. (5), this does not directly imply the respective uniqueness of its components $f$ and $g$. To see this, consider any invertible map $\Psi : \mathbb{R}^d \to \mathbb{R}^d$ and write

$$f_{\text{causal}} \circ g_{\text{causal}}^{-1} = \underbrace{f_{\text{causal}} \circ \Psi^{-1}}_{:= \hat{f}} \circ \underbrace{\Psi \circ g_{\text{causal}}^{-1}}_{:= \hat{g}^{-1}}.$$

Thus, the tuple $(\hat{f}, \hat{g})$ with $\hat{f} := f_{\text{causal}} \circ \Psi^{-1}$ and $\hat{g}^{-1} := \Psi \circ g_{\text{causal}}^{-1}$ also gives rise to the solution $h_{\text{causal}} = f_{\text{causal}} \circ g_{\text{causal}}^{-1} = \hat{f} \circ \hat{g}^{-1}$ of Eq. (5).

Our goal is to recover $g_{\text{causal}}^{-1}$ up to the equivalence class described in Definition 1, but we see that $\hat{g}^{-1} = \Psi \circ g_{\text{causal}}^{-1}$ is a solution to our considered problem, where $\Psi$ can be *any* invertible transformation. We therefore conclude that our considered problem setting is underconstrained and we cannot identify $g_{\text{causal}}^{-1}$ without additional assumptions. □

This result is unsurprising, given that we add a degree of freedom to the original problem in Eq. (3), in the form of the function $g$, without adding further constraints. As a result our problem becomes overparametrized and we can no longer uniquely recover both functions $f_{\text{causal}}$ and $g_{\text{causal}}$.

Note that if our goal is to find a predictive model that maps observations $\mathbf{X}$ to a target $Y$, which is robust to distribution shifts, this impossibility result is not an issue, since only the *composition* $(f \circ g^{-1})$ matters to solve Eq. (5). Similar findings are shown in (Arjovsky et al., 2020), underlining that predicting optimally under distribution shift does not require the causal representation.

### 4.3 Necessity of additional assumptions

So far, we have not imposed any parametric constraints on $f_{\text{causal}}$ or $g_{\text{causal}}$. The impossibility result described in Theorem 1 however implies that we require further assumptions to make progress towards identifying the latent causal variables. Since we want our results to hold for general $g_{\text{causal}}$, we refrain from beginning with assumptions on the representation function. Rather, we will investigate how parametric assumptions on $f_{\text{causal}}$ may be used to constrain class of functions $g$ that solve Eq. (5).

Notice that the functions that solve Eq. (5), $\hat{f} = f_{\text{causal}} \circ \Psi^{-1}$ and $\hat{g}^{-1} = \Psi \circ g_{\text{causal}}^{-1}$, cannot be chosen independently of each other, but are connected via the map $\Psi$. This connection motivates our reasoning behind constraining $f_{\text{causal}}$: for certain parametric choices of $f_{\text{causal}}$ (and accordingly $\hat{f}$) perhaps only a constrained set of maps $\Psi$ admits a solution to Eq. (5). If this is the case and we effectively constrain $\Psi$, we might be able to find a constraint on $f_{\text{causal}}$ such that only transformations of the form $\Psi = \mathbf{DP}$, where $\mathbf{D}$ is a diagonal matrix and $\mathbf{P}$ is a permutation matrix, which would result in recovering the causally disentangled $\hat{g}^{-1} = \mathbf{DP} g_{\text{causal}}^{-1}$, according to Definition 1.

**Linear causal mechanism.** A first natural assumption to impose is linearity of $f_{\text{causal}}$, and correspondingly of $\hat{f}$, in the hopes of constraining $\Psi$ to be a linear invertible map. This would allow us to make substantial progress towards recovering the causal variables up to permutation and rescaling, by first recovering the ground truth representation up to linear equivalence and then employing tactics to undo this linear mixing, following a common approach in causal representation learning (Ahuja et al., 2023; Zhang et al., 2023; Lachapelle et al., 2023).

We explore the implications of assuming linearity of $f_{\text{causal}}$ with an illustrative example. Assume $Y := \theta_1 Z_1 + \theta_2 Z_2 + \varepsilon_Y$, i.e. the two variable case where $d = 2$. Since we assume $f_{\text{causal}}$ to be linear, we also constrain the search space of $\hat{f}$ to the class of linear functions. Recall that $\hat{f} := f_{\text{causal}} \circ \Psi^{-1}$. Does the assumed linearity of $\hat{f}$ and $f_{\text{causal}}$ now constrain $\Psi$ to also be a linear map? Writing out

$$\hat{f} = f_{\text{causal}} \circ \Psi^{-1} = \theta_1 \psi_1^{-1} + \theta_2 \psi_2^{-1},$$

where $\psi_i^{-1}$ denotes the $i$th component of $\Psi^{-1}$, we see that e.g. by choosing $\theta_1 = 1$, $\theta_2 = 0$ only $\psi_1^{-1}$ is constrained to be linear, while $\psi_2^{-1}$ remains wholly unconstrained. We generalize this result to arbitrary choices of $\boldsymbol{\theta}$ in the following theorem.

**Theorem 2.** *Assume that $\mathbf{Z}, \mathbf{X}, Y, f_{\text{causal}}, g_{\text{causal}}$ follow the definitions in Section 2 with $d \geq 2$ and that additionally $0 \neq f_{\text{causal}} : \mathbb{R}^d \to \mathbb{R}$ is a linear function. Then, there exists an invertible nonlinear function $\Psi : \mathbb{R}^d \to \mathbb{R}^d$ such that $\hat{f} = f_{\text{causal}} \circ \Psi$ is linear. Moreover, this function $\Psi$ is not unique and can be chosen arbitrarily (that is, only constrained to be invertible) on a $d-1$-dimensional subspace of $\mathbb{R}^d$. In particular, the tuple $(\hat{f}, \hat{g})$ with $\hat{g} := g_{\text{causal}} \circ \Psi$ gives rise to the solution $h_{causal}$ of Eq. (5) in that $h_{causal} = \hat{f} \circ \hat{g}^{-1}$.*

*Proof.* We begin with an example which will later be generalized. Our goal is to elucidate if assuming $\hat{f}$ and $f_{\text{causal}}$ to be linear functions necessarily constrains $\Psi$ to be a linear map, where $\hat{f} := f_{\text{causal}} \circ \Psi^{-1}$. We

write $f_{\text{causal}}(\mathbf{z}) = \boldsymbol{\theta}^T \mathbf{z}$, where $\boldsymbol{\theta} \in \mathbb{R}^d$ and consider the case where $\boldsymbol{\theta} = (1, 0, \dots, 0)^T$. Consider any arbitrary invertible map $\Psi : \mathbb{R}^d \to \mathbb{R}^d$ with components $\psi_i$. We now write $\hat{f}$ as

$$\hat{f} = f_{\text{causal}} \circ \Psi^{-1} = \theta_1 \psi_1^{-1} + \dots + \theta_d \psi_d^{-1} = \psi_1^{-1}.$$

For this particular choice of $\boldsymbol{\theta}$, constraining $\hat{f}$ to be linear amounts to constraining $\psi_1^{-1}$ to be linear, while the other components $\psi_i^{-1}; i \in [d] \setminus \{1\}$ are not constrained at all and can be chosen arbitratrily as long as $\Psi$ remains invertible. For example, the map $\Psi(\mathbf{z}) = (z_1, z_2^3, \dots, z_d^3)$ does the job.

For a general choice of $\boldsymbol{\theta}$ we can always find an orthonormal transformation $\mathbf{A} : \mathbb{R}^d \to \mathbb{R}^d$ that maps $(\theta_1, \dots, \theta_d)^T \mapsto (1, 0, \dots, 0)^T$. Consider a nonlinear map $\Psi_0$ for the initial case $\boldsymbol{\theta} = (1, 0, \dots, 0)^T$, whose first component is linear. Define $\Psi := \mathbf{A}^T \circ \Psi_0$ which remains nonlinear. We can now write

$$\hat{f}(\mathbf{z}) := (f_{\text{causal}} \circ \Psi)(\mathbf{z}) = \boldsymbol{\theta}^T \left( \mathbf{A}^T \Psi_0(\mathbf{z}) \right) = (\mathbf{A}\boldsymbol{\theta})^T \Psi_0(\mathbf{z}) = (1, 0, \dots, 0)^T \Psi_0(\mathbf{z})$$

Thus $\hat{f}$ is linear while $\Psi$ was only constrained to be linear in its first component. $\qquad\square$

Through the generalized counterexample presented in the proof of Theorem 2, we see that the linearity requirement on $f_{\text{causal}} \circ \Psi^{-1}$ only constrains one dimension of $\Psi$, namely the one orthogonal to the kernel of $f_{\text{causal}}$. We would need one such constraint for each dimension of $\Psi$—possibly by assuming $\dim(Y) \geq d$ as in (Lachapelle et al., 2023)—in order to draw the desired linearity conclusion.

Unfortunately, the—arguably strong—assumptions presented in this section still do not suffice to recover $g_{\text{causal}}$ up to the desired equivalence class. In the following sections, we delineate possible steps forward, both in light of the presented impossibility results, as well as practical considerations.

### 4.4 Practical considerations

In the preceding sections we have shown the impossibility of learning causal representations by means of exploiting invariance, even if we consider strong parametric assumptions and an idealized setting with infinite data and perfect optimization. While these assumptions are commonplace in theoretical studies, we would like to draw particular focus to the assumption of observing all possible interventions.

The minimax problem in Lemma 1, which is the starting point of our analysis, considers the supremum over all possible interventions, i.e. all possible targets with arbitrary strength. This amounts to observing infinitely many interventional environments, which is clearly not possible in any practical scenario. Rather, one would have access to a finite number of environments during training, over which we formulate our invariance condition in hopes of generalizing to unseen environments.

Training a model on finite support, i.e. on a finite number of environments, and generalizing outside of this support amounts to extrapolation. As Christiansen et al. (2022) show, learning such *extrapolating* nonlinear functions from data with bounded support necessarily requires strong assumptions on the function class. If we do not constrain $g_{\text{causal}}$, even if $f_{\text{causal}}$ is linear, the function $f_{\text{causal}} \circ g_{\text{causal}}^{-1}$, that should generalize in the aforementioned sense, is still generally nonlinear.

If, for the sake of argument, we assume invariance as an inductive bias leads to identifiability of the underlying causal variables, we would still remain with the fundamental issue of learning such representations from finite datasets in practice. Beyond the theoretical limitations of exploiting invariance as a learning signal for causal representation learning, we are also left with the aforementioned practical ones. This gives additional weight to the conclusion that the problem setting considered in this work is highly challenging. We reflect these challenges in light of potential steps forward in the following, final section.

## 5 Discussion

Given the theoretical impossibility results outlined in Sections 4.2 and 4.3 along with the practical limitations of learning nonlinear functions that generalize outside of the training data detailed in Section 4.4, the framework of utilizing invariance as a learning signal for causal representations displays an apparent mismatch

between assumptions and the defined goal of identifiability. This misalignment compels us to reconsider either the assumptions we start from, or the goal we aim to ultimately reach. In the following we discuss both.

**Assumptions.** There is ample evidence that establishing theoretical results in the context of distributional robustness problems in conjunction with representation learning and out-of-distribution prediction hinges on some type of linearity assumption (Peters et al., 2016; Arjovsky et al., 2020; Rosenfeld et al., 2020; Krueger et al., 2021; Eastwood et al., 2022; Lachapelle et al., 2023). As we outline in Section 4.4, assuming linearity of $f_{\text{causal}}$ alone does not suffice, but perhaps additionally assuming a linear mixing function $g_{\text{causal}}$ may be sufficient for identifiability. While directly assuming linearity of $g_{\text{causal}}$ is a strong assumption, we argue that such a setting still bears practical merit. A number of works yield linearly mixed representations (Roeder et al., 2021; Ahuja et al., 2023; Lachapelle et al., 2023; Saengkyongam et al., 2023), any of which can serve as the starting point for a method that considers linear mixing functions.

We stress however, that assuming linearity of $g_{\text{causal}}$ does not directly lead to identifiability of the latent variables, as this does not alleviate the invariance of the solution of Eq. (5) to reparametrizations by an invertible map $\Psi$. Such a linear setting might allow us to utilize ideas from other works that also deal with linear mixing of the causal variables (Zhang et al., 2023; Squires et al., 2023; Buchholz et al., 2024; Varıcı et al., 2023; Lachapelle et al., 2023). Which role invariance still plays in this setting can be questioned, since these works all show identifiability of the causal variables without the enforcement of any kind of predictive robustness. E.g., by assuming $\dim(Y) \geq d$ one could directly use the results of Lachapelle et al. (2023) to achieve identifiability, albeit without exploiting any kind of invariance or interventional data, which is the core motivation of this study. We speculate that invariance doesn't hurt in these settings, but it is not immediately clear how it helps, either.

**Identifiability.** Besides discussing additional, stronger assumptions, the alternative takeaway from the results presented in this work is to reflect more broadly on the goal we have set out to accomplish. With some exceptions, element-wise identification of unobserved causal variables is the shared goal of any causal representation learning approach. While this notion of identifiability provides a common framework, it should not be viewed as purely self-serving. Causal representation learning is motivated by the promise of combining the merits of classical causal inference and modern machine learning, in order to overcome some of their respective shortcomings (Schölkopf et al., 2021).

One of the main hopes rested on causal representations is that they might lead to more robust machine learning algorithms by virtue of the invariance property inherent to causal models (cf. Section 3). This work explored if this expected property can be made explicit as a learning signal to obtain causal representations from observed mixtures. As we have shown, even in the presence of strong assumptions, invariance as an inductive bias does not lead to identifiability of the causal variables in the commonly understood sense. Fundamentally, this is due to the fact that other representations, in addition to the causal one, allow for the formulation of an invariant predictor. If we propose to use causal representations to obtain predictive robustness in the first place, this begs the question if weaker notions of identifiability than the element-wise definition used in this work suffice for practical reasons.

First works make the idea of weaker notions of identifiabiliy explicit, such as Lachapelle et al. (2022; 2024), or Saengkyongam et al. (2023) who show downstream advantages such as learning the effect of interventions that extrapolate beyond the support of the training data, even when the underlying representation is only identified up to affine transformations. Bühlmann (2018) even proposes an alternative view which suggests to define models that exhibit predictive invariance as causal. Weaker notions of identifiability exist, such as partial disentanglement introduced by Lachapelle et al. (2024), but we can also think about relaxing our definition of invariance. Starting from weaker notions of invariance that don't hold globally, but only within some region around the training support, such as those described by Rothenhäusler et al. (2021), might allow us to move towards a more pragmatic understanding of identifiability.

## 5.1 Conclusion

We have presented a theoretical investigation into the potential of predictive invariance, characteristic to causal models, as a signal for representation learning. We proposed a formal framework that allows us

to approach this question and provide first impossibility results demarcating the strength of necessary assumptions towards identifiability. Additionally, we point out practical learning issues that compound the theoretical difficulties we establish before. Given these theoretical and practical challenges, we end with a discussion reflecting on the mismatch between the assumptions and inductive biases we have chosen and the common notion of identifiability in causal representation learning. Beyond ever stronger assumptions, that must always be justified, we propose to consider less stringent notions of identifiability that still provide downstream advantages, while potentially being more permissive in attaining in practice.

## Acknowledgements

This work received funding from the European Research Council (ERC) Starting Grant CausalEarth under the European Union's Horizon 2020 research and innovation program (Grant Agreement No. 948112). S.B. received support from the German Academic Scholarship Foundation.

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

# A Proof of Lemma 2

**Lemma 2.** *Assume the data generating process in Section 2 and consider the optimization problem described in Eq. (5). Let $\mathcal{Q}^{(\mathrm{do})} := \big\{ P_{\mathbf{a},[d]}^{(\mathrm{do})}; \ \mathbf{a} \in \mathbb{R}^d \big\}$, i.e. the set of do-interventions on* all *underlying variables $\mathbf{Z}$, except $Y$, with arbitrary strength. Define $h := f \circ g^{-1} : \mathbb{R}^p \to \mathbb{R}$ and let $\mathrm{Im}(\cdot)$ denote the image of a function. Then, the composed function $h_{\mathrm{causal}} := (f_{\mathrm{causal}} \circ g_{\mathrm{causal}}^{-1})$ is the unique optimizer of Eq. (5) on $\mathrm{Im}(g_{\mathrm{causal}})$, i.e.*

$$h_{\mathrm{causal}} = (f_{\mathrm{causal}} \circ g_{\mathrm{causal}}^{-1}) \in \arg\min_{h} \ \sup_{Q \in \mathcal{Q}^{(\mathrm{do})}} \mathbb{E}_{(\mathbf{Z},Y)\sim Q} \left[ (Y - h(\mathbf{X}))^2 \right],$$

*and any other minimizer $h'$ satisfies $h' = h_{\mathrm{causal}}$ on $\mathrm{Im}(g_{\mathrm{causal}})$.*

*The same results holds true if one replaces $\mathcal{Q}^{(\mathrm{do})}$ with the set of all do-interventions $\mathcal{Q}'^{(\mathrm{do})}$ on all possible* subsets *of variables, except $Y$, with arbitrary strength.*

*Proof.* This proof largely follows the proof of Lemma 1.

Consider the decomposition of the objective

$$
\begin{aligned}
\mathbb{E}_Q \big[ (Y - h(\mathbf{X}))^2 \big] =\ & \mathbb{E}_Q \big[ (Y - h(g_{\mathrm{causal}}(\mathbf{Z})))^2 \big] \\
=\ & \mathbb{E}_Q \big[ (f_{\mathrm{causal}}(\mathbf{Z}) - h(g_{\mathrm{causal}}(\mathbf{Z})) + \varepsilon_Y)^2 \big] \\
=\ & \mathbb{E}_Q \big[ (f_{\mathrm{causal}}(\mathbf{Z}) - h(g_{\mathrm{causal}}(\mathbf{Z})))^2 \big] \\
& + \mathbb{E}_Q \big[ \varepsilon_Y^2 \big] \\
& + 2\mathbb{E}_Q \big[ (f_{\mathrm{causal}}(\mathbf{Z}) - h(g_{\mathrm{causal}}(\mathbf{Z})))\varepsilon_Y \big].
\end{aligned}
\tag{6}
$$

Again, for any interventional distribution $Q \in \mathcal{Q}^{(\mathrm{do})}$ we see that

$$
\begin{aligned}
2\mathbb{E}_Q \big[ (f_{\mathrm{causal}}(\mathbf{Z}) - h(g_{\mathrm{causal}}(\mathbf{Z})))\varepsilon_Y \big] &= 2(f_{\mathrm{causal}}(\mathbf{a}) - h(g_{\mathrm{causal}}(\mathbf{a})))\mathbb{E}_Q \big[ \varepsilon_Y \big] \\
&= 0,
\end{aligned}
$$

where we use the fact that $\varepsilon_Y$ has mean zero. Hence, for any choice of $h$ the supremum of the l.h.s. of Eq. (6) is always larger or equal to $\mathbb{E}_Q[\varepsilon_Y^2] = Var(\varepsilon_Y)$. Since $\mathcal{Q}^{(\mathrm{do})} \subseteq \mathcal{Q}'^{(\mathrm{do})}$, the supremum over $\mathcal{Q}'^{(\mathrm{do})}$ of the l.h.s. of Eq. (6) is also larger or equal to $Var(\varepsilon_Y)$.

Focusing our attention on the first term of the decomposition presented above, suppose $h \neq h_{\mathrm{causal}}$, i.e. there exists a choice of $\mathbf{b}$ s.t. $h(\mathbf{b}) \neq h_{\mathrm{causal}}(\mathbf{b})$. Recall that $h$ takes $\mathbf{X}$ as its argument, but we cannot directly intervene on $\mathbf{X} = g_{\mathrm{causal}}(\mathbf{Z})$, only on $\mathbf{Z}$.

Therefore, assume $\mathbf{b} \in \mathrm{Im}(g_{\mathrm{causal}})$ and consider an interventional distribution $Q \in \mathcal{Q}^{(\mathrm{do})}$ where we choose $\mathbf{a}$ s.t. $\mathbf{b} = g_{\mathrm{causal}}(\mathbf{a})$. Then

$$
\begin{aligned}
\mathbb{E}_Q \big[ (f_{\mathrm{causal}}(\mathbf{Z}) - h(g_{\mathrm{causal}}(\mathbf{Z})))^2 \big] &= \mathbb{E}_Q \big[ (f_{\mathrm{causal}}(\mathbf{a}) - h(\mathbf{b}))^2 \big] \\
&= \mathbb{E}_Q \big[ (h_{\mathrm{causal}}(g_{\mathrm{causal}}(\mathbf{a})) - h(\mathbf{b}))^2 \big] \\
&= \mathbb{E}_Q \big[ (h_{\mathrm{causal}}(\mathbf{b}) - h(\mathbf{b}))^2 \big] \\
&> 0,
\end{aligned}
$$

where we used $f_{\mathrm{causal}} = h_{\mathrm{causal}} \circ g_{\mathrm{causal}}$ in the second line. Again, since $\mathcal{Q}^{(\mathrm{do})} \subseteq \mathcal{Q}'^{(\mathrm{do})}$ the same statement holds for any $Q \in \mathcal{Q}'^{(\mathrm{do})}$.

Conversely, for any function $h$ that coincides with $h_{\mathrm{causal}}$ on $\mathrm{Im}(g_{\mathrm{causal}})$ the inequality above becomes an equality with zero, rendering any such function an optimizer for the considered problem. $\square$

# B Detailed related work

**Invariance, distributional robustness and causality.** As detailed in the main text, the principle of invariance is closely linked to ideas from causality. The first work that proposed to use this invariance principle

to learn causal structures from data—and kicked off a range of subsequent works—was Invariant Causal Prediction (ICP) by Peters et al. (2016), where the fact that a predictive model conditioned on all parents of a target $Y$ is invariant under interventions is used to find the parents of said target. Pfister et al. (2019) show the merit of using invariance as a signal for selecting robust prediction models for real-world biological data. Magliacane et al. (2018) utilize invariance and the Joint Causal Inference framework (Mooij et al., 2020) to find features that lead to transferable predictions across contexts, without relying on knowledge of the causal graph or specific types of interventions. Bühlmann (2018) and Meinshausen (2018) connect causality and distributional robustness and propose an alternative view where causal structures and models are defined as those that induce invariance. In Anchor Regression, Rothenhäusler et al. (2021) propose a regression model that solves a distributional robustness problem by enforcing invariance to a specific type of shift interventions. Christiansen et al. (2022) characterize various distributional robustness problems, thereby investigating the influence of the functional class of the prediction model and whether interventions extend the support of training data.

**OOD generalization.** Distributional robustness problems such as Eqs. (3) and (5) are closely related to out-of-distribution (OOD) generalization in machine learning. Rojas-Carulla et al. (2018) show that predictors that use the direct causes of a target are optimal for certain OOD problem settings. Another line of works, Risk Extrapolation (REx) (Krueger et al., 2021) and Quantile Risk Minimization (QRM) (Eastwood et al., 2022), assume a slightly different type of invariance, namely that of the risk $\mathcal{R} := \mathbb{E}[\ell(Y, \mathbf{X})]$, where $\ell$ is some loss function. REx proposes to extrapolate the convex hull of risks encountered in training to achieve robustness in test time. QRM also posits invariance of risks, but assumes a probabilistic point of view and aims not to find worst-case predictors, but those that perform well with high probability. Under additonal technical assumptions, both approaches prove that they can recover the causes of a target $Y$ in a linear SCM, if all causal variables are directly observed.

Invariant Risk Minimization (IRM) (Arjovsky et al., 2020) is another approach to OOD generalization, specifically geared towards formalizing this problem in the context of machine learning. As such, the authors consider a similar setting to ours, where we assume some underlying latent variables that permit the formulation of an invariant predictor, together with a function that maps these latents to the observations we have access to. IRM however is not interested in representation learning, as we are, but is solely aimed at learning invariant prediction models. Beyond showing, for the linear case, that IRM can separate the part of the representation that permits an invariant predictor from those parts that do not, the authors do not provide theoretical results pertaining to the identification of the underlying variables. For the nonlinear setting, follow up works have demonstrated that IRM is ill-equipped at learning predictors that perform OOD generalization (Kamath et al., 2021; Rosenfeld et al., 2020), echoing the considerations we bring forth in Section 4.4.

**Causal representation learning.** Initiated by the famous impossibility result of nonlinear independent component analysis (ICA) (Hyvärinen & Pajunen, 1999), unsupervised representation learning has been shown to be too underconstrained to be solved without additional inductive biases (Locatello et al., 2019). Later works on the identifiability of ICA problems have been able to overcome this initial obstacle by exploiting various types of auxiliary assumptions (Ilin & Honkela, 2004; Hälvä & Hyvärinen, 2020; Gresele et al., 2021; Buchholz et al., 2022; Lachapelle et al., 2022; Morioka & Hyvärinen, 2023), and more recent works in causal representation have followed this example, too. Often, the assumption of heterogeneity of the observed data distribution is made, e.g. arising from counterfactual pairs (Brehmer et al., 2022) or interventions, specifically hard do-interventions as in (Ahuja et al., 2023) or soft interventions as in (Zhang et al., 2023; Liu et al., 2022; 2023). Other approaches shift their focus to the mixing function that transforms the underlying causal variables, with one family of works focusing on how to deal with linear mixtures. Squires et al. (2023) consider linear mixtures of linear SCMs, Buchholz et al. (2024) generalize this result to nonparametric SCMs, and Varıcı et al. (2023) focus on learning the representation of linear mixtures of causal variables via a score based approach. Alternatively, some works look to exploit temporal structures, such as Lippe et al. (2022b;a; 2023) who use knowledge of interventions to achieve identifiability, Yao et al. (2021; 2022) who exploit nonstationarity or Lachapelle et al. (2024) who use an assumption about the sparsity of the SCM that generates the data.

Another approach that exploits a specific sparsity assumption is given by Lachapelle et al. (2023). Similar to our setting, the synergy of representation learning with a prediction problem is explored. Instead of considering interventions that induce heterogeneity in the data, this framework assumes multiple prediction tasks where a single, underlying representation contains the potential predictors for each individual task. This setting alone only suffices to recover the latent variables up to linear mixing, but by imposing an additional sparsity constraint, the latents are shown to be causally disentangled (cf. Definition 1).

Works considering latent DAG structure learning are also in spirit related to our setting, as they commonly assume to observe at least some nodes within the graph they aim to learn, similar to how we assume observations of the target $Y$. The main assumption for most works in this setting is the pure children assumption (Silva et al., 2006), or some variation thereof. This assumption postulates that all observed variables have only a single latent variable as a parent. Cai et al. (2019) deal with linearly transformed latent variables that have two pure children, subsequently generalized by Xie et al. (2020) to permit more than two pure children per latent in the form of the so-called Generalized Independent Noise condition. The same authors' later work further generalizes this setting to facilitate the learning of hierarchical latent variable models (Xie et al., 2022). While our approach also assumes the observation of an effect of the underlying SCM in the form of $Y$, the main difference to the aforementioned works lies in the modelling choice of the remaining observed variables, $\mathbf{X}$. In our case, we consider $\mathbf{X}$ to come from an injective transformation of the underlying variables $\mathbf{Z}$, and therefore not to be causal variables of any SCM directly, while the methods mentioned above consider all observed variables $\mathbf{X}$ to be part of the underlying, latent SCM and consequently model their relation to the latent variables $\mathbf{Z}$ in terms of surjective causal mechanisms. Given this difference, we do not require assumptions on the mechanism of $Y$ similar to those in (Xie et al., 2020) and related works pertaining to the number of observed children of latents $\mathbf{Z}$ or the linearity of the transform that yields $\mathbf{X}$.

