# OpenReview forum: "Invariance & Causal Representation Learning: Prospects and Limitations"
_TMLR — Accepted by TMLR_

### Review · Reviewer_YPxW · 2024-06-24

**Summary Of Contributions:**

In this paper, the authors investigate the problem of causal representation learning with the principle of invariance causal representation. Specifically, the authors show that under some conditions like interventions and linear causal mechanisms, the latent variables can be identified under permutation and linear transform.

**Audience:**

Yes

**Broader Impact Concerns:**

N.A.

**Claims And Evidence:**

Yes

**Requested Changes:**

1. The authors should provide a proof sketch to show how the assumptions help the identifiability of latent variables.
2. It is suggested that the authors should compare the existing methods of causal representation learning from the perspective of assumptions and conclusions.
3. The authors should prove some experiment results on synthetic and real-world datasets.

**Strengths And Weaknesses:**

However, there are some problems to solve:
1.	In Equation (1), there might be something wrong of the factorization of the joint distribution of P(Z,Y). P(Y|Z) should exist on the right-hand side of Equation (1).
2.	Although the authors illustrate the identifiability result under permutation and linear transform, the assumptions it requires are too strong. Specifically, the proposed method requires additive noise assumption, while [1] requires independent noise assumption. The proposed method requires interventional distribution of latent variables, while [3][4] does not require intervention or only soft interventions[5]. The proposed method requires a linear causal mechanism, but [6] has addressed the nonlinear cases.
3.	It is suggested that the authors should illustrate the intuition of how these assumptions benefit the identifiability of latent variables.
4.	The authors should prove some experiment results on synthetic and real-world datasets.

[1] Yao, Weiran, Guangyi Chen, and Kun Zhang. "Temporally disentangled representation learning." Advances in Neural Information Processing Systems 35 (2022): 26492-26503.
[2] Varıcı, Burak, et al. "Linear Causal Representation Learning from Unknown Multi-node Interventions." arXiv preprint arXiv:2406.05937 (2024).
[3] Xu, Danru, et al. "A sparsity principle for partially observable causal representation learning." arXiv preprint arXiv:2403.08335 (2024).
[4] Khemakhem, Ilyes, et al. "Variational autoencoders and nonlinear ica: A unifying framework." International Conference on Artificial Intelligence and Statistics. PMLR, 2020.
[5] Varici, Burak, et al. "General identifiability and achievability for causal representation learning." International Conference on Artificial Intelligence and Statistics. PMLR, 2024
[6] Kong, Lingjing, et al. "Partial disentanglement for domain adaptation." International conference on machine learning. PMLR, 2022.

---

> ### Author Response · Authors · 2024-07-08
> **Reply to Reviewer YPxW**
>
> We thank the reviewer for taking the time to review our manuscript and providing valuable feedback.
> As a general remark, we would like to point out what we believe is a misunderstanding regarding our main result. We do not provide an identifiability result, but rather an impossibility result. We discuss your further comments and concerns in light of this, below:
>
> 1. $P(Y|Z)$ is indeed on the r.h.s. of Eq. 1, as we point out that $Z_{d+1} = Y$ in the paragraph preceding this equation.
> 2. As mentioned, we do not illustrate or prove any form of identifiability, instead we discuss impossibility results. Furthermore, additive and independent noise are not mutually exclusive, in fact we also assume independent noise terms. The fact that we begin from the strict assumption of additive noise rather strengthens our impossibility result. We include interventional data by necessity, since invariance is precisely defined around it (see e.g. invariant causal prediction [1]).
> Very many causal representation learning (CRL) approaches include interventional data in some way [2-7], and using hard interventions is simply for ease of analysis. Our impossibility result shows that even when considering the strongest assumptions of hard interventions, we cannot make progress with invariance in CRL. Soft interventions (which are a generalization of hard interventions) should therefore not be expected to help us either.
> We only consider a linear mechanism for a single causal variable (the target $Y$) and here also only as a first attempt to obtain identifiability because the nonlinear case is underconstrained, as we show in Thm. 1.
> As other reviewers also raise the point that our assumptions should be further discussed and placed in better context, we will add a clarified discussion of them to the camera ready version. See also the following point for details in this regard.
> 3. One of the key motivations behind learning truly causal representation is to exploit the invariance property inherent to causal models. Our study sets out to probe if directly regularizing for invariance allows us to arrive at such representations. To begin our analysis, we consider a setting where invariance has proven to be useful for causal inference [1, 8], albeit without the added challenge of representation learning. Our main finding is that even in such a setting, where we have arguably strong assumptions (linearity, additive noise, hard interventions), the chosen constraints do not suffice to achieve identifiability as commonly defined in the literature. Given this result, we do not expect less strong assumptions to provide a benefit, either. As we detail in the Discussion section, (predictive) invariance does not readily lend itself to the identifiability of causal representations. However, we believe our study provides an impulse to reconsider the commonly shared goal of identifiability, and help move towards a more pragmatic goal for CRL. E.g., if the main application for the learned representation is robust prediction, identifying the unique causal representation may be over-specific, and it may suffice to identify the underlying variables up to bijection.
> 4. Ideally, we would have also liked to validate our theoretical results experimentally. However, our main theoretical results are on the impossibility of identifying latent representations using invariance as an inductive bias. Therefore, we would also not expect experiments to provide any further insights, other than that they would fail.

---

> > ### Author Response · Authors · 2024-07-08
> > **References**
> >
> > [1] Jonas Peters, Peter Bühlmann, and Nicolai Meinshausen. Causal inference by using invariant prediction:identification and confidence intervals. Journal of the Royal Statistical Society. Series B (Statistical Methodology), 78(5):947–1012, 2016.
> >
> > [2] Kartik Ahuja, Divyat Mahajan, Yixin Wang, and Yoshua Bengio. Interventional Causal Representation Learning. In Proceedings of the 40th International Conference on Machine Learning, pp. 372–407, 2023.
> >
> > [3] Jiaqi Zhang, Kristjan Greenewald, Chandler Squires, Akash Srivastava, Karthikeyan Shanmugam, and Caroline Uhler. Identifiability Guarantees for Causal Disentanglement from Soft Interventions. In Advances in Neural Information Processing Systems, volume 36, pp. 50254–50292, 2023.
> >
> > [4] Chandler Squires, Anna Seigal, Salil S. Bhate, and Caroline Uhler. Linear Causal Disentanglement via Interventions. In Proceedings of the 40th International Conference on Machine Learning, pp. 32540–32560, 2023.
> >
> > [5] Simon Buchholz, Goutham Rajendran, Elan Rosenfeld, Bryon Aragam, Bernhard Schölkopf, and Pradeep Kumar Ravikumar. Learning Linear Causal Representations from Interventions under General Nonlinear Mixing. In Advances in Neural Information Processing Systems, 37, 2023.
> >
> > [6] Sébastien Lachapelle, Pau Rodríguez López, Yash Sharma, Katie Everett, Rémi Le Priol, Alexandre Lacoste, and Simon Lacoste-Julien. Nonparametric Partial Disentanglement via Mechanism Sparsity: Sparse Actions, Interventions and Sparse Temporal Dependencies. arXiv preprint arXiv:2401.04890, 2024.
> >
> > [7] Phillip Lippe, Sara Magliacane, Sindy Löwe, Yuki M. Asano, Taco Cohen, and Efstratios Gavves. CITRIS: Causal Identifiability from Temporal Intervened Sequences. In Proceedings of the 39th International Conference on Machine Learning, pp. 13557–13603, 2022.
> >
> > [8] Nicolai Meinshausen. Causality from a Distributional Robustness Point of View. In Proceedings of the IEEE Data Science Workshop (DSW 2018), pp. 6–10, 2018.

---

> > > ### Comment · Reviewer_YPxW · 2024-08-28
> > >
> > > Thanks for the authors. They addressed my concerns

---

### Review · Reviewer_T5kx · 2024-06-27

**Summary Of Contributions:**

1. This work aims to address a challenging fundamental theory in the field of causal representations, specifically the identifiability of causal representation learning.

2. Beyond the commonly-used assumption of having only observed data x, this study introduces an additional observed variable y, in the hope that y can assist in the identifiability of causal representation.

3. This work attempts to utilize the introduced variable y through the principle of independent causal mechanisms, a common assumption in causality.

The most interesting aspect of this work, in my opinion, is the introduction of another variable y to assist in the identifiability of causal representation, going beyond the traditional setting of using only observed data. Although the introduction of the new variable may require additional labeling for the data, it is interesting to see how the new variable can reduce the need for strong assumptions on x (as most works do), given the fact that identifiability is nearly impossible without such assumptions.

**Audience:**

Yes

**Broader Impact Concerns:**

None,

**Claims And Evidence:**

No

**Requested Changes:**

1) My first concern is the assumptions used in this work. There are many assumptions presented in Section 2, Problem Setting. For example, there are assumptions such as additive noise, injectivity of the mapping from latent space to x, and zero-mean noise related to y. These assumptions are introduced in various places throughout the paper, making the context of the proposed theorem unclear. As a result, it is difficult to judge the main results of the proposed theorem.

2)  There are also some ambiguous assumptions. The work assumes the observation of (X,Y) across multiple environments, but it is unclear how many environments are needed in the proposed theories. The work also defines subsets S where the underlying latent variables Z have undergone an intervention in each environment. What is the size of these subsets S? Different sizes may imply different requirements for the data. In general, we might prefer the size to be small, which corresponds to the sparse mechanism shift assumption. Additionally, do you need to know the number of latent variables?

3) The main result in this work, in my opinion, is the connection between the solution uniqueness in Eq. (3) or (5) and the independent causal mechanism. Generally, the result requires data from various environments where interventions on all possible subsets of variables must occur. This result confuses me. Currently, many works are related to such environments, and generally, I think that once we obtain data from various environments with interventions on each variable, identifiability can be achieved from x alone [1-4]. If we can obtain such intervened data, why do we need to introduce another variable y? Does the introduction of y reduce the assumptions required in previous works?

4) Given the above, I think it is crucial to conduct a thorough comparison of the assumptions used in this work with those in existing literature. The comparison should not be limited to hard interventions; recent papers using soft interventions should also be considered, for example, [5-7]. In fact, I think this might be as a good motivation for this study.

5) I attempted to understand the target variable y, but found it challenging, particularly in general application scenarios such as labeling image data in classification tasks. I think it would be beneficial to further clarify the meaning of the target variable y, especially considering that the work may impose certain assumptions on y.


6) While I understand that this work may prioritize 'discussion', conducting experiments to validate the claims in the proposed theorems could significantly enhance its credibility. I also acknowledge that the fundamental results may rely on strong assumptions, which could make real-world experiments challenging. Nevertheless, providing simulations or even simplified experiments would be beneficial.


[1] Ahuja, Kartik, et al. "Interventional causal representation learning." International conference on machine learning. PMLR, 2023.

[2] von Kügelgen, Julius, et al. "Nonparametric identifiability of causal representations from unknown interventions." Advances in Neural Information Processing Systems 36 (2024).

[3] Squires, Chandler, et al. "Linear causal disentanglement via interventions." International Conference on Machine Learning. PMLR, 2023.

[4] Jiang, Yibo, and Bryon Aragam. "Learning nonparametric latent causal graphs with unknown interventions." Advances in Neural Information Processing Systems 36 (2024).

[5] Liu, Yuhang, et al. "Identifying weight-variant latent causal models." arXiv preprint arXiv:2208.14153 (2022).

[6] Zhang, Jiaqi, et al. "Identifiability guarantees for causal disentanglement from soft interventions." Advances in Neural Information Processing Systems 36 (2024).

[7] Liu, Y., Zhang, Z., Gong, D., Gong, M., Huang, B., Hengel, A. V. D., ... & Shi, J. Q. (2023). Identifiable Latent Polynomial Causal Models Through the Lens of Change. ICLR 2024

**Strengths And Weaknesses:**

Strengths: see above,
Weaknesses: see below.

---

> ### Author Response · Authors · 2024-07-09
> **Reply to Reviewer T5kx [1/2]**
>
> Thank you very much for your time and for providing such detailed comments. We are encouraged by the fact that you recognize our considered problem to be challenging and interesting. Please find our detailed response below:
>
> 1. Other reviewers also raised the point that the assumptions deserve better context and presentation, which we will do in the updated version. Generally speaking, our main assumptions are drawn from the most closely related works which we build upon. The injectivity of the mapping from $\mathbf{Z}$ to $\mathbf{X}$ is ubiquitous in causal representation learning (CRL) and used in almost all related works [1-6]. Additive noise is similarly assumed in [7, 8], which serves as the starting point for our study.  The noise on $Y$ is assumed to have zero mean for ease of theoretical analysis, as we could also consider non-zero mean noise, subtract the mean from $Y$, and end up in the setting we directly assume.
> 2. One of the original motivations for our work was the hope that invariance might reduce the number of required environments compared to other interventional CRL approaches. We do not have a definite answer on this yet because identifiability is not even possible under the strong assumption of having access to all possible (hard)-interventional environments. Therefore, we unfortunately also cannot provide something like a lower bound.
> We also want to emphasize that our results don’t require interventions on _all possible_ subsets of variables —- our paper just says that our results hold in this case. Note that our results also hold if we just consider interventions on all variables (except $Y$) at once. We considered these two cases for stringency between [9] and our work, however we can also remove this detail if it causes confusion.
> Also note that the considered interventional environments come from the distributional robustness literature [7,8,10], as opposed to the interventional CRL literature [1-6]. In the latter case, one hopes to minimize the number of such environments, while in the former one considers the “worst case” of all possible interventions with arbitrary strength to show that the causal predictor $f_\text{causal}$ is _maximally_ robust (i.e. invariant) under any considerable intervention. This explains where the set $\mathcal{Q}$ comes from in our Eqs. 3 and 5. As we point out in Section 4.4, this causes problems in practice, which would still persist even if identifiability were achievable theoretically.
> Our results do not require us to know the number of latent variables.
> Also see the next point for further discussion.
> 3. Whether or not including Y allows reducing assumptions compared to previous works summarizes the original motivation for our study:
> Learning _causal_ representations in ML models is often motivated by predictive invariance/OOD prediction [11]. Given the usefulness of invariance for causal inference (think of independent mechanisms, for example) where the variables of interest are directly observed (so no representations involved) [7-9], we set out to investigate whether or not invariance provides a benefit in the context of CRL, as this has not explicitly been studied before. Since we consider _predictive_ invariance, we necessarily need to include a target variable $Y$, in line with all prior works that our study builds upon [7-9]. As we show in Eqs. 3 and 5, invariance permits environments where more than one variable is intervened upon in a single environment. Thus, we hypothesize that we might use interventional data in a more general way than previous CRL approaches that only allow single node interventions.
> Since we show that invariance does not suffice to obtain identifiability (as it is defined in [1-6]), it appears that using invariance does not provide much of an advantage over other works that consider interventional environments. However, as we outline in the Discussion section, there is another avenue for future work. Clearly, the assumptions we consider are not strong enough to achieve identifiability. However, we believe we should not search for ever stronger assumptions, but rather reconsider the notion of identifiability that we aim to achieve. We could imagine more pragmatic definitions of identifiability that focus on learning an invariant causal mechanism for a pre-defined prediction task, together with the representations of its causal parents. In spirit with the suggestions brought forward in [10], future work could consider a mechanism and representation identified precisely when it is invariant in our desired sense.

---

> > ### Author Response · Authors · 2024-07-09
> > **Reply to Reviewer T5kx [2/2]**
> >
> > 4. Following your remark, as well as other reviewers’ comments, we will provide an updated overview of related works. We will also discuss works that use soft rather than hard interventions. We decided to start with hard interventions in hopes of deriving generalized results for soft interventions, similar to how [2] generalizes [1]. But given that we obtain an impossibility result with hard interventions, we do not expect soft ones to provide any benefit or different outcome.
> > 5. For applications, all we require is that there is an observed target $Y$ that is the effect of some unobserved upstream variables $\mathbf{Z}$ and that we have access to a (high dimensional) mapping $\mathbf{X} = g(\mathbf{Z})$. With this in mind, one application could be where $Y$ is a measurable bio-medical state like heart rate or blood pressure, while $\mathbf{Z}$ are latent physiological variables describing the state of different organ systems or cell states, but we only measure higher dim. mappings of these latent variables via medical imaging tools. Another example is in climate science, where $Y$ can be a directly measurable variable such as global mean temperature, $\mathbf{Z}$ are high-level climatic modes of variability like the North Atlantic Oscillation or El Niño and $\mathbf{X}$ are high-dimensional satellite measurements. In both cases, we can assume that $\mathbf{Z}$ causes $Y$, but we cannot observe $\mathbf{Z}$ directly.
> > Recently, such predictive settings with a target variable have been considered in CRL, see also [12,13].
> > 6. Since we cannot prove identifiability under the considered setting, there is no direct method to be derived from our results and consequently no experimental validation to perform. We would only expect any given approach to fail in identifying the latent variables, which does not add further insights in our opinion.
> >
> > ### References
> > [1] Kartik Ahuja, Divyat Mahajan, Yixin Wang, and Yoshua Bengio. Interventional Causal Representation Learning. In Proceedings of the 40th International Conference on Machine Learning, pp. 372–407, 2023.
> >
> > [2] Jiaqi Zhang, Kristjan Greenewald, Chandler Squires, Akash Srivastava, Karthikeyan Shanmugam, and Caroline Uhler. Identifiability Guarantees for Causal Disentanglement from Soft Interventions. In Advances in Neural Information Processing Systems, volume 36, pp. 50254–50292, 2023.
> >
> > [3] Chandler Squires, Anna Seigal, Salil S. Bhate, and Caroline Uhler. Linear Causal Disentanglement via Interventions. In Proceedings of the 40th International Conference on Machine Learning, pp. 32540–32560, 2023.
> >
> > [4] Simon Buchholz, Goutham Rajendran, Elan Rosenfeld, Bryon Aragam, Bernhard Schölkopf, and Pradeep Kumar Ravikumar. Learning Linear Causal Representations from Interventions under General Nonlinear Mixing. In Advances in Neural Information Processing Systems, 37, 2023.
> >
> > [5] Sébastien Lachapelle, Pau Rodríguez López, Yash Sharma, Katie Everett, Rémi Le Priol, Alexandre Lacoste, and Simon Lacoste-Julien. Nonparametric Partial Disentanglement via Mechanism Sparsity: Sparse Actions, Interventions and Sparse Temporal Dependencies. arXiv preprint arXiv:2401.04890, 2024.
> >
> > [6] Phillip Lippe, Sara Magliacane, Sindy Löwe, Yuki M. Asano, Taco Cohen, and Efstratios Gavves. CITRIS: Causal Identifiability from Temporal Intervened Sequences. In Proceedings of the 39th International Conference on Machine Learning, pp. 13557–13603, 2022.
> >
> > [7] Jonas Peters, Peter Bühlmann, and Nicolai Meinshausen. Causal inference by using invariant prediction:identification and confidence intervals. Journal of the Royal Statistical Society. Series B (Statistical Methodology), 78(5):947–1012, 2016.
> >
> > [8] Nicolai Meinshausen. Causality from a Distributional Robustness Point of View. In Proceedings of the IEEE Data Science Workshop (DSW 2018), pp. 6–10, 2018.
> >
> > [9] Mateo Rojas-Carulla, Bernhard Schölkopf, Richard Turner, and Jonas Peters. Invariant Models for Causal Transfer Learning. Journal of Machine Learning Research, 19(36):1–34, 2018.
> >
> > [10] Peter Bühlmann. Invariance, Causality and Robustness. arXiv preprint arXiv:1812.08233, 2018.
> >
> > [11] Bernhard Schölkopf, Francesco Locatello, Stefan Bauer, Nan Rosemary Ke, Nal Kalchbrenner, Anirudh Goyal, and Yoshua Bengio. Toward Causal Representation Learning. Proceedings of the IEEE, 109(5):612–634, 2021.
> >
> > [12] Sébastien Lachapelle, Tristan Deleu, Divyat Mahajan, Ioannis Mitliagkas, Yoshua Bengio, Simon Lacoste-Julien, and Quentin Bertrand. Synergies between Disentanglement and Sparsity: Generalization and Identifiability in Multi-Task Learning. In Proceedings of the 40th International Conference on Machine Learning, pp. 18171–18206, 2023.
> >
> > [13] Sorawit Saengkyongam, Elan Rosenfeld, Pradeep Ravikumar, Niklas Pfister, and Jonas Peters. Identifying Representations for Intervention Extrapolation. arXiv preprint arXiv:2310.04295, 2023.

---

> > > ### Comment · Reviewer_T5kx · 2024-08-08
> > >
> > > Thank you for the response. Although the non-identifiability results are not surprising, this work provides a relatively detailed discussion on the relationship between invariance and identifiability analysis in causal representation learning.
> > >
> > > I think the work could be accepted, but I would suggest adding a careful discussion on why identifiability is not even possible under the strong assumption of having access to all possible interventional environments. This result appears to be somewhat inconsistent with current findings that use interventions for identifiability.

---

### Review · Reviewer_JLTs · 2024-07-03

**Summary Of Contributions:**

This paper explores the relationship between the invariance of causal mechanisms and the identifiability of causal representation learning (CRL). It establishes an impossibility result, demonstrating that invariance alone is insufficient for achieving the desired identifiability often considered in the CRL literature.

**Audience:**

Yes

**Claims And Evidence:**

Yes

**Requested Changes:**

No major changes. Please refer to the above comments for minor ones.

**Strengths And Weaknesses:**

Strengths:

1. The problem is clearly defined, and the relationship between invariance and CRL is thoroughly explored within a formal framework. This framework is novel and has not been introduced in the literature before, making it quite interesting.

2. The writing is clear and concise.

3. The paper effectively argues that the invariance principle is significant not only in causal inference but also in CRL.

Weaknesses:

1. No identifiability results in the considered general framework are provided. While impossibility results and discussions on potential assumptions are interesting, they may not offer sufficient guidance for addressing the real problem.

2. Some relevant lines of work in CRL are not discussed or mentioned in the paper. For example, the identifiability of nonlinear ICA can be achieved without introducing auxiliary variables, through functional constraints such as PNL, conformal maps, or sparsity.

---

> ### Author Response · Authors · 2024-07-08
> **Reply to Reviewer JLTs**
>
> Thank you for reviewing our manuscript and for providing positive feedback on our work. We are encouraged by the fact that you consider our problem setting interesting, as well as our presentation to be clear and concise. Please find our replies to your raised comments below:
>
> 1. While we agree that identifiability results provide more actionable insights, we believe that our impossibility results, especially under the strong assumption of linear $f_\text{causal}$, help to gain a more clear understanding of the difficulties of this problem setting. As we outline in the Discussion section, we hope that this work can provide an impulse to rethink the commonly shared notion of identifiability in the field towards a possibly more task-focussed definition, where invariance is one of the key criteria. If robust prediction is the main task, perhaps identifying the causal representation in an element-wise fashion is over-specific and it suffices to learn the latent variables up to a bijection.
> 2. Thank you for pointing out these omissions. We will include references and discuss the mentioned works in nonlinear ICA together with the other suggestions on related works that the other reviewers have mentioned.

---

### Decision · Action_Editor_7bGk · 2024-08-29

**Recommendation:** Accept as is

**Comment:**

This paper investigates the connection between causal invariance mechanism and the identifiability of the causal representations. Through theoretical analysis, the paper gives the conclusion that invariance mechanism alone cannot imply the identifiability of causal representations. As agreed by all the reviewers, the impossibility theorem proposed in this paper provides valuable new insights to the field, and there is unanimous support for its acceptance. Based on the review comments, I also recommend acceptance.

**Audience:**

yes

**Claims And Evidence:**

yes